# Uremic Toxins and Inflammation: Metabolic Pathways Affected in Non-Dialysis-Dependent Stage 5 Chronic Kidney Disease

**DOI:** 10.3390/biomedicines12030607

**Published:** 2024-03-07

**Authors:** María Peris-Fernández, Marta Roca-Marugán, Julià L. Amengual, Ángel Balaguer-Timor, Iris Viejo-Boyano, Amparo Soldevila-Orient, Ramon Devesa-Such, Pilar Sánchez-Pérez, Julio Hernández-Jaras

**Affiliations:** 1Health Research Institute Hospital La Fe, 46026 Valencia, Spain; marta_roca@iislafe.es (M.R.-M.); hernandez_jul@gva.es (J.H.-J.); 2University and Polytechnic La Fe Hospital, 46026 Valencia, Spain; viejo_iri@gva.es (I.V.-B.); amsolde@gmail.com (A.S.-O.); devesasuch@hotmail.com (R.D.-S.); sanchez_pil@gva.es (P.S.-P.); 3Big Data AI and Biostatistics Platform, Health Research Institute Hospital La Fe, 46026 Valencia, Spain; julian_amengual@iislafe.es (J.L.A.); angel_balaguer@iislafe.es (Á.B.-T.)

**Keywords:** ACKD, inflammation, metabolomics, energy metabolism, uremic toxins

## Abstract

Chronic kidney disease (CKD) affects approximately 12% of the global population, posing a significant health threat. Inflammation plays a crucial role in the uremic phenotype of non-dialysis-dependent (NDD) stage 5 CKD, contributing to elevated cardiovascular and overall mortality in affected individuals. This study aimed to explore novel metabolic pathways in this population using semi-targeted metabolomics, which allowed us to quantify numerous metabolites with known identities before data acquisition through an in-house polar compound library. In a prospective observational design with 50 patients, blood samples collected before the initial hemodialysis session underwent liquid chromatography and high-resolution mass spectrometer analysis. Univariate (Mann–Whitney test) and multivariate (logistic regression with LASSO regularization) methods identified metabolomic variables associated with inflammation. Notably, adenosine-5′-phosphosulfate (APS), dimethylglycine, pyruvate, lactate, and 2-ketobutyric acid exhibited significant differences in the presence of inflammation. Cholic acid, homogentisic acid, and 2-phenylpropionic acid displayed opposing patterns. Multivariate analysis indicated increased inflammation risk with certain metabolites (*N*-Butyrylglycine, dimethylglycine, 2-Oxoisopentanoic acid, and pyruvate), while others (homogentisic acid, 2-Phenylpropionic acid, and 2-Methylglutaric acid) suggested decreased probability. These findings unveil potential inflammation-associated biomarkers related to defective mitochondrial fatty acid beta oxidation and branched-chain amino acid breakdown in NDD stage 5 CKD, shedding light on cellular energy production and offering insights for further clinical validation.

## 1. Introduction

CKD represents a significant public health issue due to its substantial economic impact and its role as an independent risk factor for cardiovascular diseases. It is estimated to affect 12% of the global population with a proportion of 0.1% being in stage 5 (estimated glomerular filtration rate < 15%) [1]. Despite advancements in therapies for anemia, mineral and bone disorder (MBD), and cardiovascular risks [2,3,4], mortality remains notably elevated in individuals with NDD stage 5 CKD.

The uremic phenotype, characterized by metabolic waste accumulation due to declining kidney function, leads to symptoms such as nausea, vomiting, fatigue, anorexia, mental health status changes, and others, resulting in in a reduced quality of life and increased morbidity and mortality [5]. Inflammation, a significant contributor to the uremic phenotype, also plays a role in elevated cardiovascular and overall mortality [6].

CKD acts as a mild but chronic and persistent inflammatory process akin to ‘inflammaging’, in which the tissues exhibit a low-grade, chronic, systemic inflammation, with the absence of a clear origin [7]. Studies indicate that over half of stage 3 CKD patients have elevated CRP levels with a much higher prevalence in stage 5 CKD patients and those undergoing renal replacement therapy [8]. Systemic inflammation in these groups of patients with advanced kidney disease is linked to adverse outcomes, including a reduced quality of life and increased mortality from cardiovascular disease (CVD) and infectious complications. These outcomes are also associated with acquired immune dysfunction, depression, osteoporosis, and nutritional disturbances [9].

Inflammation has a multifactorial etiology in CKD including exogenous factors (like the use of a central venous catheter and intestinal dysbiosis), tissue factors (like hypoxia and fluid and sodium overload), comorbidities (like congestive heart failure, diabetes, and malnutrition), and uremic toxins. While some medium-sized molecule toxins have been recognized as key players in this pathology, numerous metabolic pathways remain unexplored [6].

To uncover novel pathways, metabolomics may be the best suited method within the ‘omics’ sciences. This discipline employs various analytical methods to detect small molecules (metabolites) in a given sample, aiming to evaluate these substances both quantitively and qualitatively to explore their potential applications in diagnosis, treatment, and prognosis [10]. These compounds, collectively referred to as the metabolome, are shaped by both internal factors, such as interactions between the proteome and the genome, and external influences, like lifestyle, medications, and underlying diseases. The metabolome is considered the most accurate representation of an individual’s phenotype and is therefore well suited to evaluate the array of uremic toxins in the human body [11].

Targeted metabolomics has been used to assess changes in endocannabinoid and oxylipin levels in hemodialysis patients [12], relating them to their inflammatory process, and untargeted metabolomics has been used to study the changes they undergo when suffering from critical conditions [13]. Yet, our approach, utilizing semi-targeted metabolomics to assess the potential biological signature for inflammation in NDD stage 5 CKD patients through semi-targeted metabolomics, is not described in the literature.

Semi-targeted approaches bridge the gap between untargeted and targeted procedures, quantifying numerous metabolites with known identities before data acquisition, achieved through the utilization of an in-house polar compound library. Consequently, this approach enables the identification of measured metabolites in a unified workflow. In contrast, untargeted metabolomics concentrates on comprehensive detection and relative quantitation, while targeted metabolomics concentrates on the precise measurement of predefined, much smaller groups of metabolites, allowing for absolute quantitation [14].

Our goal is to delineate the metabolic profile in this cohort, emphasizing inflammation-associated metabolites as potential therapeutic targets. Addressing this could prevent cardiovascular complications and enhance the survival of this patient group [15].

## 2. Materials and Methods

### 2.1. Study Design and Participants

We aimed to examine the impact of various metabolites on inflammation in patients with NND stage 5 CKD. We established an observational, prospective unicentric study and collected clinical and analytical data before the first dialysis session.

Initially, we performed a descriptive analysis of the population to identify its characteristics. Subsequently, we delved into the metabolomic profile of the patients before their initial hemodialysis session.

The study included individuals with stage 5 CKD requiring imminent renal replacement therapy, voluntarily opting for hemodialysis at the Advanced Chronic Kidney Disease Unit at Hospital La Fe (Valencia) in alignment with the healthcare team’s counsel. Inclusion criteria included being over 18 years of age and having scheduled initiation of the hemodiafiltration technique, while exclusion criteria involved suboptimal initiation of the technique due to clinical destabilization or life expectancy of less than one year.

Additionally, we collected information on inflammation in patients based on their blood analysis C-reactive protein values (CRP). Patients with CRP values ≤ 2 mg/L were considered non-inflamed, while those with CRP > 2 mg/L were considered inflamed. Only 30% were considered non-inflamed (CRP < 2 mg/L).

We determined this cut-off point value based on the identification of CRP levels between 1 and 3 mg/L as the gray zone of inflammation, where cardiovascular risk could possibly increase [16].

All patients came to the hospital and underwent a system-based medical interview and basic physical examination before the sample extraction and first hemodialysis session to ensure that the influence of clearly infectious events was minimal.

### 2.2. Blood Sample Collection and Preparation

Blood samples were collected and stored for analysis, followed by processing through centrifugation, with the resulting serum stored at −80 °C until study completion. Multiple biochemical parameters were assessed, including CRP.

Afterwards, 180 µL of cold methanol (MeOH) was added to 20 µL of serum for protein precipitation and extraction of polar compounds.

Following double centrifugation at 13,000× *g* (10 min, 4 °C), 20 µL of the supernatant was transferred to a 96-well plate for liquid chromatography coupled to mass spectrometry (LC-MS) analysis. Then, 70 µL of H_2_O and 10 µL of internal standard mix solution (reserpine, leucine, enkephalin, phenylalanine-d5, 20 µM) were added to each sample. Quality control (QC) was prepared by mixing 10 µL from each serum sample. Blank was prepared by replacing serum with ultrapure water in order to identify potential artifacts from the tube, reagents, and other materials. Finally, plasma samples, QCs, and blanks were injected in the chromatographic system. In order to avoid intra-batch variability, as well as to enhance quality and reproducibility, the scheme analysis consisted of random injection order and analysis of QC every 6 plasma samples. Blank analysis was performed at the end of the sequence.

### 2.3. UPLC-HRMS Analysis

Samples were analyzed on an Ultra-Performance Liquid Chromatography (UPLC) system coupled to an Orbitrap QExactive spectrometer (Thermofisher, Waltham, MA, USA). The chromatographic separation was performed using an Xbridge BEH Amide column (150 × 2 mm, 2.5 µm particle size; Waters) with a 25 min runtime, an injection volume of 5 µL, and a column temperature of 25 °C. The autosampler was set to 4 °C, and a flow rate of 105 µL/min was used with water and 10 mM ammonium acetate as Mobile Phase A with and acetonitrile (ACN) as Mobile Phase B. The gradient was as follows: 0 min, 90% B, 2 min, 90% B; 3 min, 75%; 7 min, 75% B; 8 min, 70%, 9 min, 70% B; 10 min, 50% B; 12 min, 50% B; 13 min, 25% B; 14 min, 25% B; 16 min, 0% B, 20.5 min, 0% B; 21 min, 90% B; 25 min, 90% B.

The electrospray ionization was employed in both positive and negative modes (ESI +/−), in full mass acquisition, with a resolving power of 140,00 and two events: mass ranges from 70 to 700; mass ranges from 700 to 1700 Da. Data were acquired in the centroid mode.

### 2.4. Data Preprocessing

Raw data were converted to mzXML format using Mass Converter and then processed in EI-MAVEN software (V0.11.0) to generate a peak table containing *m*/*z* retention times and intensities of polar compounds. Peak areas were extracted and annotated using an in-house polar compound library. Data from positive and negative modes were merged for statistical analysis.

Before the statistical analysis, data quality (reproducibility, stability) was evaluated by means of the internal standard’s stability and the QC’s coefficients of variation (CVs). Those molecular features with CVs > 30% were removed from the data matrix, and a normalization method (LOESS) was also used to eliminate intra-batch variability due to technical differences. Finally, the filtered peak table was used for statistical analysis.

### 2.5. Statistical Analysis

First, we conducted a univariate analysis consisting of exploring which metabolomic variables were associated with the presence of an inflammatory process. To perform this, we identified which metabolites were significantly upregulated with a fold change of 1.5 in the inflammatory group compared with the non-inflammatory group using a non-parametric Mann–Whitney test (log2(fold change) = log2(MInflammation = 1/MInflammation = 0).)

With regard to the multivariable analysis, a logistic regression model, suitable for binary response variables, with the application of LASSO regularization was used. LASSO (least absolute shrinkage and selection operator) is a regularization technique that helps in both variable selection and regularization by shrinking some regression coefficients toward zero, contributing to a more interpretable model. This algorithm optimized the coefficients of the logistic regression for each variable (metabolite) to improve performance, specifically reducing classification errors. It achieved this by progressively decreasing coefficients using a multiplier parameter in the algorithm’s cost function until they equaled zero, effectively eliminating the corresponding variable. The extent of coefficient minimization and elimination was determined by an additional parameter called lambda. A higher lambda value resulted in greater coefficient penalization, reducing their values to zero and, consequently, reducing the number of variables included in the model. The optimal lambda value was determined using Cross-Validation. Different models were fitted, with one group of subjects excluded in each iteration to estimate their classification into the ‘inflamed = no’ or ‘inflamed = yes’ group. The error for each case was calculated, and the lambda value minimizing this error was considered the optimal choice. In this model, k was set to 50, meaning one patient was excluded for estimation in each iteration, with a focus on minimizing classification errors. Standard coefficients (SCs) and odds ratios (ORs) were calculated for each variable.

The sign of the coefficients indicates how changes in variables affect the probability of being inflamed or not. Positive coefficients imply a higher probability of inflammation, while negative coefficients reduce the probability. Variables with larger absolute standardized coefficients are more important in the model, reflected in their odds ratios (OR = e^coefficient). ORs represent the multiplier for the probability of belonging to the ‘inflammation = yes’ group compared to ‘inflammation = no’ assuming a one-standard-deviation increase in the variable, as we use standardized coefficients.

## 3. Results

This study employed an observational prospective design to investigate metabolite impact on inflammation in 50 NDD stage 5 CKD patients. Blood samples were collected before the start of the first hemodialysis session.

### 3.1. Demographic Characteristics of the Sample

The study cohort, with a mean age of 69.46 years (SD ± 13.28), included 38% females, and exhibited an average height of 165.18 cm (SD ± 8.83) and a mean weight of 75.51 kg (SD ± 15.76), resulting in an average BMI of 27.61 (SD = 4.71). Half of the subjects were smokers, and an equal percentage had type 2 diabetes. Hypertension was prevalent in 94% of the population, while 76% exhibited dyslipidemia.

The summary of the demographic and clinical characteristics of the sample (*n* = 50) can be found in Table 1.

A semi-targeted metabolomic approach was followed, utilizing liquid chromatography and high-resolution mass spectrometry with an in-house library of polar compounds. The results can be found in the Appendix A. Univariate (Mann–Whitney test) and multivariate (logistic regression with LASSO regularization) analyses identified metabolomic variables associated with inflammation.

### 3.2. Univariate Analysis

The median concentration of homogentisic acid (*p* = 0.0007), 3-indoxyl-sulfate (*p* = 0.0087), 2-phenylpropionic acid (*p* = 0.0351), and cholic acid (*p* = 0.0165) significantly decreased in the presence of inflammation, as indicated by a statistically significant Mann–Whitney test. The magnitude of change was greater in cholic acid, in which the median decreased by a factor of 2.79 in inflamed patients, and was lowest in homogentisic acid, in which it was decreased by a factor of only 1.60.

On the other hand, the median concentration of dimethylgycine (*p* = 0.0015), pyruvate (*p* = 0.0066), lactate (*p* = 0.0156), ketobutyric acid (*p* = 0.0458), and adenosine 5-phosphosulfate (*p* = 0.0211) significantly increased in the presence of inflammation. The magnitude of change was greater in adenosine 5-phosphosulfate, which was 0 in non-inflamed patients and almost negligible in lactate, where the median increased by a factor of 1.52 in inflamed patients.

A numerical summary is provided in Table 2, including the respective medians and interquartile ranges of the variables with significant changes between the two groups, and Figure 1 visually represents this information in a volcano plot.

### 3.3. Multivariate Analysis

*N*-Butyrylglycine (SC = 0.63, OR 1.87), dimethylglycine (SC = 0.60, OR 1.81), 2-oxoisopentanoic acid (SC = 0.56, OR 1.76), pyruvate (SC = 0.56, OR 1.76), and L-fucose (SC = 0.2 OR 1.22) show positive standard coefficients and odds ratios greater than 1, indicating a positive association with the outcome (inflammation = yes).

Meanwhile, homogentisic acid (SC = −0.33, OR 0.72), 2-phenylpropionic acid (SC = −0.26, OR 0.77), and 2-methylglutaric acid (SC = −0.26, OR 0.78) show negative standard coefficients and odds ratios lower than 1, indicating a negative association with the outcome (inflammation = yes).

Notably, uric acid (SC = 0.13, OR 1.14), adenosine 5′-phosphosulfate (SC = 0.09, OR 1.09), nicotinuric acid (SC = 0.08, OR 1.09), L-hydroxyproline (SC = 0.03, OR 1.03), and lactate (SC = 0.01, OR 1.01) have a small positive coefficient and odds ratios close to 1, indicating a minor positive influence on the outcome, whereas 3-indoxyl sulfate (SC = −0.08, OR 0.92), proline (SC = −0.07, OR 0.94), and nicotinamide (SC = −0.03, OR 0.97) have a small negative coefficient and odds ratios close to 1, in this case indicating a minor negative influence on the outcome.

Finally, we provide a numerical summary in Table 3, and in Figure 2 we graphically represent the odds ratios (ORs) obtained for each variable.

## 4. Discussion

Our goal was to describe the metabolic profile observed in NDD stage 5 CKD patients, focusing on the metabolites associated with inflammation.

Inflammation in NDD stage 5 CKD patients has received limited attention in metabolomics studies. We conducted a comprehensive literature review over the last 5 years on PubMed using the terms metabolome- AND CKD and inflammation, yielding 47 results, and the terms metabolome- AND dialysis AND inflammation, yielding 13 results. After excluding reviews and studies lacking metabolomics or other omics, 12 relevant results remained.

In 2020, a Korean group performed an untargeted metabolomic analysis to identify metabolites related to the development of CKD in the general population, such as citrulline and kynurenine, though it did not focus specifically on inflammation in CKD [17].

In the same year, a Japanese group conducted targeted metabolomics to investigate a new molecule’s response targeting the uremic toxin TMAO, which aided in reducing renal inflammation and fibrosis [18]. This metabolite also showed relevance to inflammation in peritoneal dialysis with peritoneal infection [19].

In 2021, a lipidomics study identified a therapeutic target in nephrectomized mice with CKD by inhibiting Alox15 inhibition and increasing PGD2, thereby reducing inflammation and fibrosis [20].

In 2023, a proteomics study identified 233 plasma proteins related to cardiovascular disease and inflammation, pinpointing 21 proteins associated with kidney function decline, among others, such as TNFRSF11A [21].

A 2022 study, using targeted metabolomics, found that elevated quinolinic and anthranilic acids were linked to increased cardiovascular risk in CKD [22].

Another study, published in 2020, explored the interplay between the gut microbiome and circulating metabolites in CKD, shedding light on potential links between the microbiome, inflammation, and renal health [23].

A recently published study using data from 91,532 Biobank participants explored lipid biomarkers associated with incident CKD using nuclear magnetic resonance [24]. Similarly, a 2019 study using untargeted metabolomics in 2155 participants, including stage 1–5 CKD patients and healthy controls, identified five metabolites, including 5-MTP, strongly correlated with clinical markers of kidney disease [25].

Another study focused on gut dysbiosis and proinflammatory immune signature in children with CKD. Elevated TNF alpha and sCD14 indicated inflammation, while microbiome and metabolite analyses revealed stage-dependent alterations [26].

In 2022, a study used untargeted metabolomics to explore the levels of endocannabinoids and oxylipins in female hemodialysis patients compared to healthy controls. The observed changes may contribute to inflammation and cardiovascular issues observed in hemodialysis patients [12].

Finally, in a study from 2021, serum metabolomic profiling identified predictive models for mortality in critically ill patients with acute kidney injury (AKI) requiring renal replacement therapy, indicating links to inflammatory processes and muscle wasting [27].

As observed, omics sciences have been seldom employed to specifically measure inflammation in CKD, but rather metabolites related to CKD development and in none of the cases precisely in NDD stage 5 CKD patients. None of the mentioned works observed the metabolites found in our study.

In our case, univariate and multivariate analyses identified significant associations. Adenosine-5′-phosphosulfate, dimethylglycine, and pyruvate showed an increased probability of inflammation. Conversely, cholic acid, homogentisic acid, and 2-phenylpropionic acid exhibited opposite patterns. The multivariate analysis indicated increased inflammation risk with *N*-butyrylglycine, dimethylglycine, 2-oxoisopentanoic acid, and pyruvate, while homogentisic acid, 2-phenylpropionic acid, and 2-methylglutaric acid suggested decreased probability.

If we focus on those that were associated with inflammation in both univariate and multivariate analyses, we find dimethyglycine and pyruvate.

Despite its prevalent promotion as a nutritional supplement, because of its fundamental functions in the biosynthesis of methionine and folic acid [28], dimethylgycine (DMG) is a known uremic toxin. It is linked to homocysteine metabolism through the enzyme betaine-homocysteine methyl transfuse (BHMT) that converts glycine betaine to N,N, dimethylglycine. The accumulation of DMG in CKD can lead to hyperhomocystinemia by inhibiting BHMT activity. Homocysteine, a by-product of methionine metabolism, oxidizes in plasma, generating free radicals that harm endothelial cells and promote LDL oxidation. It also elevates platelet aggregation, blood coagulation, and vascular smooth muscle growth in vitro [29]. Furthermore, dimethylglycine can induce reactive oxygen species (ROS) production in liver mitochondria through reverse electron transfer to complex I [30]. In a rat study inducing chronic inflammation through lipopolysaccharide administration, metabolic analysis revealed that dimethylglycine, among others, was related to mitochondrial dysfunction and carbon metabolism [31]. Another study investigated the association between the metabolic profile and inflammatory cytokines in new onset psoriasis, showing significant increases in DMG [32].

Pyruvate, on the other hand, is an alpha ketoacid that is key to the cellular metabolism, acting as a link between glycolysis and the tricarboxylic acid cycle (TCA) to provide energy. Once inside the mitochondria, the majority of pyruvate undergoes oxidative phosphorylation to produce ATP, but firstly it is converted to acetyl-CoA through the pyruvate dehydrogenase complex [33]. Aberrant pyruvate metabolism plays an especially prominent role in cancer, neurodegeneration, heart failure, and other conditions. For example, patients with heart failure have decreased levels of ATP and phosphocreatine levels because the phosphorylation of pyruvate is decreased, and this pattern is also observed in chronic progressive diseases such as chronic pulmonary disease (COPD), obesity, diabetes, and aging. It could be hypothesized that chronic kidney disease, which has many metabolic similarities to the aforementioned situations, could also present with aberrant pyruvate metabolism [34]. Indeed, a study performed with mice with a subtotal nephrectomy showed significant suppression of pyruvate dehydrogenase and suppression of proximal tubular mitochondrial respiration and ATP synthesis [35]. Another study performed in stage 4–5 CKD patients concluded that CKD patients had lower levels of pyruvate dehydrogenase and this correlated with muscle dysfunction measured by handgrip strength [36].

We also found that *N*-butyrilglycine, 2-oxoisopentanoic acid, and L-fucose were strongly correlated with inflammation in the multivariate analysis.

*N*-Butyrylgycine belongs to the class of compounds known as acylglycines. Acyglycines are typically found in low concentrations as by-products of fatty acid metabolism. However, individuals with certain inborn errors of metabolism, such as ethylmalonic encephalopathy, exhibit increased excretion of these compounds and the analysis of these metabolites in bodily fluids can be utilized for diagnosing disorders linked to mitochondrial fatty acid beta oxidation [37].

Acyglycines are created by the process of transesterification, where an acylcoA ester combines with glycine, and this reaction is facilitated by the enzyme acyl-COA-glycine-N-acyl-transferase, also known as glycine N-acylase. They have also been described to increase in congenital fatty oxidation diseases like SCAD (short-chain Acyl-CoA dehydrogenase deficiency), in which there is an elevated presence of butyrilglycine in urinary excretion [38]. Another fatty acid oxidation disease (FAOD), multiple acyl-coA dehydrogenase deficiency, hinders the beta oxidation of fatty acids as well as the breakdown of branched amino acids and has shown elevated plasma levels of acylglycines.

These metabolic alterations can lead to impaired gluconeogenesis, ureagenesis, and ketogenesis, resulting in hypoglycemia, lactic acidemia, and hyperammonemia, which can be accompanied by hepatomegaly, steatosis, and reduced ketone production. Muscle and cardiac tissues are notably affected due to their high energy demand from fatty acid oxidation [38].

Therefore, although the association of this metabolite with inflammation was not previously described in either the general population or CKD, it would make sense for it to be related to inflammation in CKD patients through the affected organs and pathways.

It is interesting to highlight the relationship between the metabolites butyrylglycine, dimethylglycine, and pyruvate (Figure 3). Butyrylglycine, an acylglycine, is considered a by-product of faulty fatty acid beta oxidation in which, due to a defect, acetyl-coA is not correctly produced for entry into the Krebs cycle and energy production. On the other hand, the increased levels of pyruvate and its connection with inflammation also may indicate an inability to produce acetyl-CoA and, consequently, energy. Finally, dimethylglycine is linked to butyrylglycine due to its association with glycine, although its link to inflammation is attributed to its relationship with homocysteine.

2-Oxoisopentanoic acid, also known as alpha-ketoisovaleric acid, is an abnormal metabolite (a branched chain ketoacid) resulting from the incomplete breakdown of branched chain amino acids, specifically valine (as shown in Figure 4). It acts as a neurotoxin and an acidogen. It is classified as metabotoxin and its elevated levels can lead to chronic health issues. This is what happens in maple syrup urine disease [37]. BCKAs disrupt brain energy metabolism by inhibiting crucial enzymes such as pyruvate dehydrogenase, alpha-ketoglutarate dehydrogenase, and mitochondrial respiration in rat models [39]. Furthermore, it has been observed that BCKA excess may induce mitochondrial oxidative stress, redox imbalance, and cytokine release in macrophages of type 2 diabetes patients [40]. Once more, this study’s findings reveal a compromised energy metabolism.

L-fucose is a hexose deoxy sugar and is a common component of many glycans and glycolipids produced by mammalian cells [37]. As a free sugar, it is usually present at very low concentrations in humans, but elevated levels have been reported in the past in breast cancer, ovarian cancer, lung cancer, and liver cancer [41,42,43].

However, at the present time, its exogenous administration was also identified via a neural network model using the Drug Signatures Database as a candidate for treating severe inflammation in IgAN by reducing the deposition of complement C3 [44], and its administration was also found to alleviate intestinal epithelial damage via upregulating FUT2, impeding oxidative stress, mitochondrial dysfunction, and cell apoptosis [45]. In kidney transplants, it has been studied as a decoy molecule, significantly reducing ischemia reperfusion injury [46]. This phenomenon could be explained by considering the substance as one that naturally elevates in the context of inflammation, but whose exogenous administration has anti-inflammatory properties.

Finally, adenosine 5′-phosphosulfate (APS) was strongly correlated with inflammation in the univariate analysis. It is a type of purine ribonucleoside monophosphate, which is a nucleotide containing a purine base attached to a ribose with one monophosphate group [37]. It is a precursor in the biosynthesis of 3′-phosphoadenosine-5′-phosphosulfate (PAPS), which, in turn, is a universal sulfuryl donor in various biochemical pathways. This reaction is catalyzed by the enzyme 3′-phosphoadenosine-5′-phosphosulfate synthase (PAPSS), which has two human isoforms in humans, PAPSS 1 and PAPPS2, the latter of which is expressed predominantly in the liver, cartilage, and adrenal glands [47].

When PAPSS2 is deficient, the conversion of APS to PAPS is impaired, leading to an accumulation of APS which can disrupt the normal sulfation processes in cells. It has been described that this deficiency can lead to osteochondrodysplasias [48].

Most recently, it has been described that PAPSS2 is decreased in colon cancers in mice and humans, correlating with a worse survival due to an increased intestinal permeability and bacteria infiltration [49].

Therefore, we may hypothesize that an increase in APS may be related to a decrease in PAPSS2, with the elevated inflammation observed being associated with increased intestinal permeability and bacterial infiltration.

However, several influential variables explaining the lack of inflammation were also detected. In the case of homogentisic acid (OR 0.71) and phenylpropionic acid (OR 0.77), the changes were significant in quantity and statistically significant in both univariate and multivariate analyses. Nevertheless, not all of them made sense at a pathophysiological level.

Homogentisic acid, an intermediate product of the tyrosine and phenylalanine metabolism, can act as an osteotoxin and a renal toxin when present at high levels. Elevated levels are linked to alkaptonuria, a rare genetic disorder. Mutations in homogentisate 1,2-dioxygenase can cause its accumulation, resulting in dark urine, cartilage damage, kidney stones, and heart valve issues [50]. There is no existing literature that describes it as protective against inflammation, kidney disease, or cardiovascular disease. We conducted a thorough verification of the correct metabolite annotation and it seems the most likely candidate in terms of formula and retention time. Considering the hypothesis-generating nature of untargeted metabolomics, the actual relationship of this metabolite with inflammation should be tested.

2-Phenylpropionic acid, on the other hand, has been found to be produced by various kinds of bacteria, such as Acinetobacter, Bacteroides, Bifidobacterium, Clostridium, Enterococcus, Escherichia, Eubacterium, Klebsiella, Lactobacillus, Pseudomonas, and Staphylococcus [37]. In a recent study conducted by Cho et al. in mice, it was discovered that mice exhibiting higher levels of phenylpropionic acids had a decreased susceptibility to acetaminophen-induced hepatotoxicity and CYP2E1-mediated hepatotoxicity by carbon tetrachloride [51]. This kind of hepatotoxicity appears when acetaminophen, carbon tetrachloride, and other compounds are converted to reactive metabolites, which can directly react with proteins introducing alterations in protein structure or folding. Subsequently, these proteins are processed by antigen presenting cells appearing ‘foreign’ to the immune system, leading to an immune response [52]. Therefore, it would make sense that high levels of 2-phenylpropionic acid are associated with a decrease in the risk of presenting an inflammatory profile in our patients.

Also correlated with the lack of inflammation in the multivariate analysis, we found 2-Methylglutaric acid.

2-Methylglutaric acid belongs to the class of organic compounds known as methyl-branched fatty acids. These are fatty acids with an acyl chain that has a methyl branch. They are mainly produced by bacteria in the guts of ruminant animals and ingested by humans through dairy products [53]. Other methyl-branched fatty acids, such as phytanic acid, have been shown to have favorable effects on glucose metabolism, energy expenditure, and immune functions when present in naturally occurring amounts. However, they could be toxic when found in excess (e.g., Refsum disease) [54].

And, finally, in the univariate analysis, cholic acid (*p* = 0.02) was correlated with lack of inflammation.

Along with chenodeoxycholic acid, it constitutes one of the two major bile acids produced by the liver, derived from cholesterol. This compound is able to activate the farnesoid X receptor (FXR), which has a regulatory effect on bile acid production by repressing de novo bile acid synthesis. However, FXR also has a role in the regulation of lipids and glucose metabolism. It is able to activate reverse cholesterol transportation, preventing the development of atherosclerosis, and participates in fatty acid (FA) oxidation, preventing FA-induced lipotoxicity [55]. This is important as dyslipidemia and altered lipid metabolism are often overlooked as risk factors in NDD CKD patients compared to non-CKD patients and cardiovascular risk in this population linearly correlates to low-density-lipoprotein (LDL) cholesterol levels [56].

TGR5, another important bile acid receptor, promotes mitochondrial fission and induces thermogenic activity in adipocytes, thus preventing obesity and insulin resistance in mice. Although chenodeoxycholic acid and deoxycholic acid showed a higher potency in activating FXR and TGR5, they were regrettably not measured in our library [45] (p. 2).

These results reveal possible biomarkers associated with inflammation that are linked to impaired mitochondrial fatty acid beta oxidation and the breakdown of branched-chain amino acids in NDD stage 5 CKD, providing insights into cellular energy production and suggesting avenues for additional clinical validation.

Firstly, there is a need to validate the metabolites identified in our study through targeted metabolomics, allowing for the establishment of their exact concentrations and determining their significance in the alteration of each metabolic pathway. Secondly, it would be interesting to explore metabolites that our results suggest may be related but were not found in the polar compound in-house library, such as other acylglycines, chenodeoxycolic acid, and deoxycholic acid (in relation to FXR) or other branched-chain ketoacids. Finally, considering the importance of fatty acid metabolism in the inflammation of NDD stage 5 CKD patients, lipidomics could provide valuable insights in this field.

This study has several limitations. Firstly, as we mentioned in the previous paragraph, our data only reveal trends or changes in response intensities, not exact concentrations. Secondly, we decided on the use of an in-house library of polar compounds in order to limit non relevant results and simplify interpretation, but this may have avoided the finding of several metabolites. Lastly, unfortunately, due to our time and budget constraints, we were only able to include 50 patients, so multiple comparisons and subgroup analyses were clearly limited. It would be specially interesting to examine the population with diabetic kidney disease given its notably higher risk of cardiovascular events, CKD progression, and disability which are commonly associated with inflammation. The identification of biomarkers and therapeutic targets could help individualize the treatment and prognosis of each patient [57].

## 5. Conclusions

In conclusion, inflammation in NDD stage 5 chronic kidney disease leads to premature aging, a decrease in the quality of life, and an increased cardiovascular risk. Despite having multiple etiologies, the pathophysiology underlying this condition is not entirely clear, and we do not have specific biomarkers or targets in this particular group of patients. Therefore, our study, through semi-targeted metabolomics, highlights metabolic pathways that contribute significantly to it, such as mitochondrial fatty acid beta oxidation or the incomplete breakdown of branched-chain amino acids, impacting cellular energy production. These findings provide a starting point for subsequent targeted metabolomics or lipidomics studies that can validate and expand the insights presented in this research.

## Figures and Tables

**Figure 1 biomedicines-12-00607-f001:**
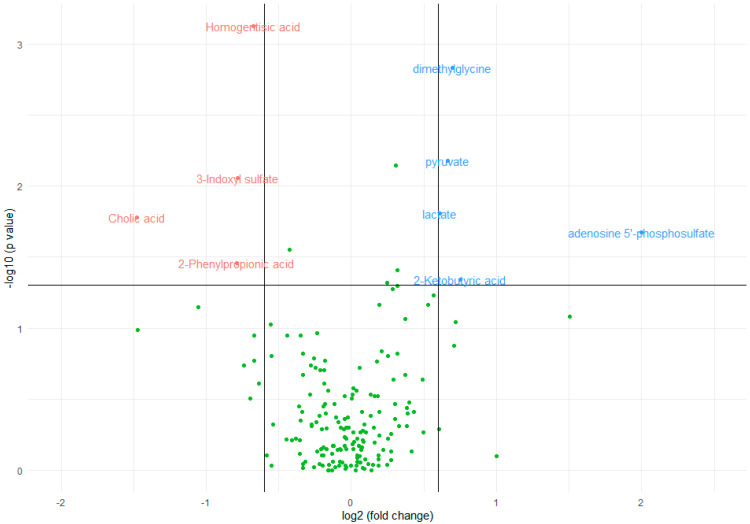
Volcano plot of metabolomic variables. The horizontal black line represents the limit of statistical significance; values above it indicate variables with significantly different medians. The vertical black lines represent a log2(fold change) of −0.6 and +0.6. Values above +0.6 represent variables with a median for inflammation = yes that is 1.5 times greater in magnitude than for inflammation = no (in blue). Values below −0.6 represent variables with a median for inflammation = no that is 1.5 times greater in magnitude than for inflammation = yes (in red). The green dots represent metabolites whose changes in relation to inflammation are not clinically or statistically significant.

**Figure 2 biomedicines-12-00607-f002:**
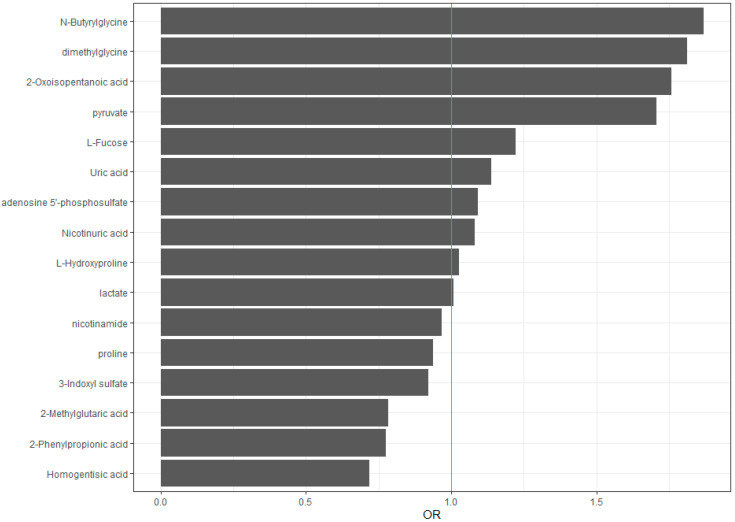
Graphical representation of the odds ratios (ORs) obtained with the logistic–LASSO model. ORs close to 1 (red vertical line) indicate variables with a lesser effect on the model.

**Figure 3 biomedicines-12-00607-f003:**
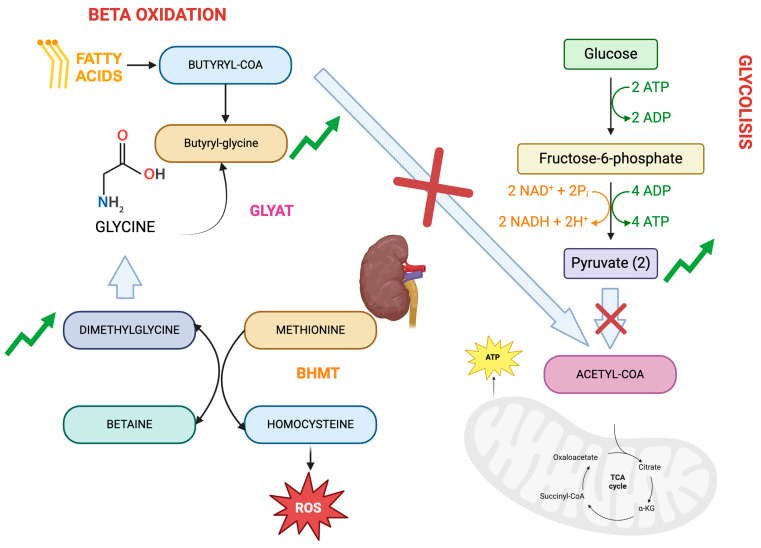
Graphical representation of the metabolic relationship between butyrylglycine, dimethylglycine, and pyruvate.

**Figure 4 biomedicines-12-00607-f004:**
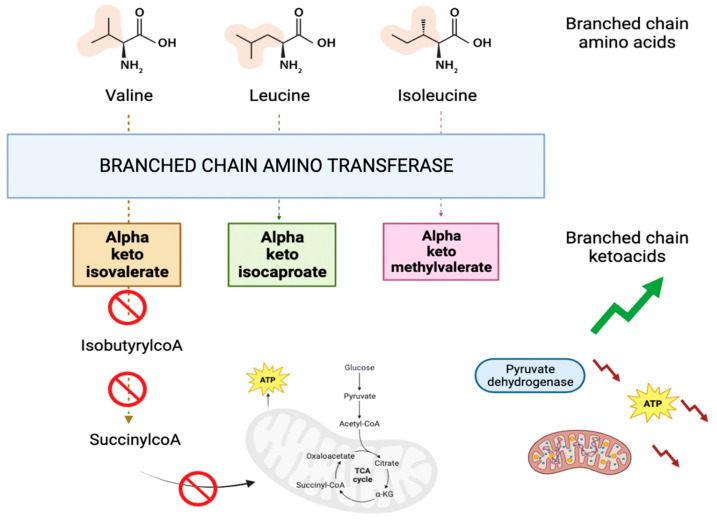
Graphical representation of the metabolic pathway involving alpha keto isovalerate.

**Table 1 biomedicines-12-00607-t001:** Demographic characteristics of the sample.

Variable	Mean/Frequency
Age (years) (mean ± SD)	69.46 ± 13.28
Female gender, n (%)	19 (38%)
Height (cm) (mean ± SD)	165.18 ± 8.83
Weight (kg) (mean ± SD)	75.51 ± 15.76
BMI (mean ± SD)	27.61 ± 4.71
Smoker, n (%)	25 (50%)
Type 2 diabetes, n (%)	25 (50%)
Hypertension, n (%)	47 (94%)
Dyslipidemia, n (%)	38 (76%)
Urea (mg/dL) (mean ± SD)	163.12 ± 45.50
Creatinine (mg/dL) (mean ± SD)	5.78 ± 1.11
eGFR (mL/min) (mean ± SD)	9.17 ± 1.76
Uric acid (mg/dL) (mean ± SD)	6.52 ± 1.78
Corrected calcium for serum albumin (mg/dL) (mean ± SD)	8.90 ± 0.77
Phosphate (mg/dL) (mean ± SD)	4.99 ± 1.43
Sodium (mEq/L) (mean ± SD)	139.88 ± 2.86
Potassium (mEq/L) (mean ± SD)	4.65 ± 0.68
Chloride (mEq/L) (mean ± SD)	105.96 ± 4.25
CRP (mg/dL) (mean ± SD)	11.98 ± 21.38
Hemoglobin (g/dL) (mean ± SD)	10.60 ± 1.36
Leukocytes (×10^9^/L) (mean ± SD)	7.33 ± 2.51
Platelets (×10^9^/L) (mean ± SD)	180.00 ± 63.84

**Table 2 biomedicines-12-00607-t002:** Numerical summary of metabolomic variables with significant differences in their medians based on inflammation = no and inflammation = yes.

Variable	Median Inflammation = No	IQR Inflammation = No	Median Inflammation = Yes	IQR Inflammation = Yes	Mann–Whitney Statistic	*p* Value
Homogentisic acid	658,062	340,396	412,251	244,065	417	0.0007
dimethylglycine	125,416	69,804	203,207	189,224	116	0.0015
pyruvate	71,565,240	20,799,310	113,500,840	111,279,800	136	0.0066
2-Oxoisopentanoic acid	10,934,585	2,820,924	13,506,204	3,898,370	137	0.0071
3-Indoxyl sulfate	207,231,408	124,405,200	120,266,360	89,655,840	385	0.0087
lactate	80,909,480	44,669,740	123,676,000	58,251,360	149	0.0156
Cholic acid	84,849	181,404	30,452	59,027	375	0.0165
adenosine 5′-phosphosulfate	0	0	33309	76,439	162	0.0211
Kynurenic acid	791,827	358,109	590,577	401,863	366	0.0281
2-Phenylpropionic acid	424,658	237,796	246,110	303,895	362	0.0351
Uric acid	9,967,433	3,054,808	12,411,336	4,516,135	165	0.0391
2-Ketobutyric acid	663,583	806,580	1,117,340	859,427	168	0.0458
L-Hydroxyproline	719,050	233,060	851,651	257,328	169	0.0482

**Table 3 biomedicines-12-00607-t003:** Variables, standardized coefficients, and odds ratios obtained (LASSO).

Variables	Standard Coefficients	Odds Ratios
N-Butyrylglycine	0.6252470	1.8687074
dimethylglycine	0.5952781	1.8135351
2-Oxoisopentanoic acid	0.5640526	1.7577816
pyruvate	0.5341265	1.7059575
Homogentisic acid	−0.3305017	0.7185631
2-Phenylpropionic acid	−0.2552945	0.7746883
2-Methylglutaric acid	−0.2450682	0.7826511
L-Fucose	0.2005541	1.2220797
Uric acid	0.1293206	1.1380549
adenosine 5′-phosphosulfate	0.0880096	1.0919986
3-Indoxyl sulfate	−0.0827405	0.9205900
Nicotinuric acid	0.0795591	1.0828095
proline	−0.0650603	0.9370110
nicotinamide	−0.0329016	0.9676337
L-Hydroxyproline	0.0277108	1.0280983
lactate	0.0097718	1.0098197

## Data Availability

The data on which the article is based are available as Appendix A.

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
