# Peer review of "Uremic Toxins and Inflammation: Metabolic Pathways Affected in Non-Dialysis-Dependent Stage 5 Chronic Kidney Disease"

_biomedicines, 2024, doi:10.3390/biomedicines12030607_

Round 1

Reviewer 1 Report

Comments and Suggestions for Authors

The manuscript thoroughly investigates the metabolic profile linked to inflammation in non-dialysis-dependent stage 5 CKD patients. The authors extensively discuss the pathophysiological implications of identified metabolites, particularly exploring how changes in dimethylglycine, pyruvate, and others may contribute to inflammation. Despite some limitations, the detailed analyses and interpretations establish a solid groundwork for future research in this domain. I consider the paper valuable, and while there are no major concerns regarding its scientific soundness, addressing one significant comment and making a few minor improvements could further enhance its overall quality.

Major comment:

The criteria for classifying patients as inflammatory are unclear in the study. Please provide clarification on why a CRP value > 2 mg/dl was chosen as the inflammation cutoff point. If this decision was based on the median CRP value, please specify. Additionally, consider discussing whether GFR levels within the 5 CKD stage or comorbidities, such as diabetes or hypertension, were taken into account and how they might impact the observed metabolic changes. Understanding these interactions could contribute valuable insights into the context of inflammation in the patient cohort.

Manor comments:

1)    The manuscript utilizes diverse terms such as "CKD patients," "stage 5 patients," and "stage 5 patients with kidney disease," “patients with stage 5 renal insufficiency” to refer to the patient cohort. For improved clarity and coherence, it is recommended to consistently employ a single term in both the title and text.

2)    I propose acknowledging the specific context of advanced stages of CKD in relation to inflammation in the introduction. This acknowledgment will assist readers in comprehending the study's significance within the distinct subgroup of patients under investigation.

3)    Move the descriptive analysis of the patient cohort from the Methods section to the Results section, providing additional details to characterize the patient population. Consider incorporating routine clinical data (blood pressure, serum creatinine, GFR, hemoglobin, CRP, electrolytes, etc.), and ensure the inclusion of column titles in Table 1 for improved clarity.

4)    Omit the term "p-value" from the textual description, retaining only the numeric representation (e.g., p=0.001). Additionally, replace commas with decimal points in the format (e.g., p=0.001 instead of p=0,001).

5)    Provide the results of logistic regression in the standard format of Odds Ratios (OR) with corresponding 95% Confidence Intervals (CI) to facilitate a clearer understanding of the statistical significance of the findings. Avoid duplicating statistical information present in both the text and the tables.

Comments on the Quality of English Language

There are areas where minor improvements could be made for clarity and coherence.

Author Response

Please, see attachment. 

Reviewer 2 Report

Comments and Suggestions for Authors

The subject is interesting and metabolomic analysis in pre-dialysis patients carries the potential of identifying new players adding to the long list of uremic toxins responsible for inflammation and its clinical consequences.

The description of methodology is clear and convincing, the results are presented in a concise way. The discussion contains valuable criticism and balance between scientific findings and careful interpretation of their potential clinical meaning.

Minor

Abstract contains the statement about semi-targeted metabolomics – please specify.

It ends up with the statement about insights for further clinical validation – such subject should be mentioned in the discussion, suggesting perspectives for future investigation.

Discussion contains review-like elements, including figures. In order to keep the balance, it should be supplemented with more focus on comparison with other Authors’ results. If there are none, their uniqueness should be underlined. Any similar results concerning e.g. patients in earlier stages of CKD or already on dialysis?

Conclusions should be rephrased – they mainly contain repetition of the results.

Author Response

Please, see attachment

Reviewer 3 Report

Comments and Suggestions for Authors

Fernandez and Colleagues Explored, in this originale article, the mebolomic profile of CKD stage 5 patients. 
The topic is of interest in my opinion. Moreover, the statistical approach used is adequate.  
I have two major concerns:

1) Did the authors stratify patients for PCR or high sensitivity PCR to assess infiammation? It is important, to gain correct results, to split patients correctly. Can you try to use IL-6 values?

2) it is also important to adjust the analysis for potential confounders of infiammation, such as clinical relevant events (infectious, clear inflammatory events). Did you consider this?

3) please expand a bit the discussion by citing papers discussing the importance of improve prognosis of CKD patients such as doi: 10.3390/life12081202.  and doi: 10.1016/j.numecd.2015.04.001.  among others. 

Comments on the Quality of English Language

this is fine

Author Response

Please, see attachment

Round 2

Reviewer 3 Report

Comments and Suggestions for Authors

I think the article was improved.